# Reliability and Validity of the ONAPS Physical Activity Questionnaire in Assessing Physical Activity and Sedentary Behavior in French Adults

**DOI:** 10.3390/ijerph18115643

**Published:** 2021-05-25

**Authors:** Marc Charles, David Thivel, Julien Verney, Laurie Isacco, Pauliina Husu, Henri Vähä-Ypyä, Tommi Vasankari, Michèle Tardieu, Alicia Fillon, Pauline Genin, Benjamin Larras, Bruno Chabanas, Bruno Pereira, Martine Duclos

**Affiliations:** 1Département de Santé Publique, Centre Hospitalier Universitaire Gabriel Montpied, 63003 Clermont-Ferrand, France; bruno.chabanas@uca.fr; 2Laboratoire des Adaptations Métaboliques à l’Exercice en Condition Physiologique et Pathologique (AME2P), Université Clermont Auvergne, 63178 Aubière, France; david.thivel@uca.fr (D.T.); julien.verney@uca.fr (J.V.); laurie.isacco@uca.fr (L.I.); 3The UKK Institute for Health Promotion Research, 3500 Tampere, Finland; pauliina.Husu@ukkinstituutti.fi (P.H.); henri.Vaha-Ypya@ukkinstituutti.fi (H.V.-Y.); tommi.vasankari@ukkinstituutti.fi (T.V.); 4Faculty of Medicine and Health Technology, Tampere University, 33014 Tampere, Finland; 5Observatoire National de l’Activité Physique et de la Sédentarité (ONAPS), Faculté de Médecine, Université Clermont Auvergne, 63000 Clermont-Ferrand, France; m.tardieu@onaps.fr (M.T.); a.fillon@onaps.fr (A.F.); p.genin@onaps.fr (P.G.); b.larras@onaps.fr (B.L.); 6Unité de Biostatistique, Direction de la Recherche Clinique et de l’Innovation, Centre Hospitalier Universitaire Gabriel Montpied, 63003 Clermont-Ferrand, France; bpereira@chu-clermontferrand.fr; 7Service de Médecine du Sport et des Explorations Fonctionnelles, Centre Hospitalier Universitaire Gabriel Montpied, 63003 Clermont-Ferrand, France; mduclos@chu-clermontferrand.fr

**Keywords:** criterion validity, concurrent validity, accelerometry, self-report, test–retest study

## Abstract

This study was conducted to assess the validity and reliability of a new questionnaire, the ONAPS-PAQ, developed to assess physical activity (PA) and sedentary behaviors (SB) in the general population. A total of 137 healthy adults aged 18 to 69 years were included. Following completion of two physical activity questionnaires (ONAPS-PAQ and GPAQ, the Global physical activity questionnaire) to study concurrent validity, participants wore an accelerometer (UKK-RM42) for 7 days to study criterion validity. A subsample (*n* = 36) also completed a 7-day-interval test–retest protocol to assess its reliability. Reliability was tested by the intraclass correlation coefficient (ICC) and Kappa coefficient; concurrent and criterion validity by the Spearman correlation coefficient (ρ) and Bland-Altman plot analyses. The ONAPS-PAQ showed good reliability (ICC = 0.71–0.98; Kappa = 0.61–0.99) and concurrent validity (ρ = 0.56–0.86), but only poor criterion validity (ρ = 0.26–0.41), and wide limits of agreement. Self-reported and accelerometer-measured SB were better correlated with ONAPS-PAQ than GPAQ (0.41 vs. 0.26, respectively) and medians were comparable, whereas the GPAQ underestimated SB (SB_acc_ = 481 (432–566), SB_ONAPS_ = 480 (360–652), SB_GPAQ_ = 360 (240–540) min·day^−1^; *median* (*q1*-*q3*)). The ONAPS-PAQ provides good reliability and acceptable validity for the measurement of PA and SB and seems to provide a better assessment of SB than GPAQ.

## 1. Introduction

Physical inactivity and sedentary behavior are major risk factors for chronic disease and obesity, generating premature death as well as a heavy social and economic burden [1,2,3,4]. Regular physical activity is now clearly recognized for its protective effect for non-communicable diseases, associated with reduced mortality [5,6,7], improved quality of life [8], and presents therapeutic effects for many chronic diseases [9]. Considering the constant rise in physical inactivity and sedentary behaviors in the last 50 years [10,11], an increase should, unfortunately, continue regarding their current prevalence in youth [12]; properly monitoring physical activity (PA) and sedentary behavior (SB) is clearly of public health interest.

Physical activity questionnaires (PAQs) are often the most feasible method to assess PA in large-scale studies and are widely used in surveillance systems for risk stratification and when examining the etiology of disease in large observational studies. Although they are prone to measurement error and bias due to misreporting, either deliberate (social desirability bias) or because of cognitive limitations related to recall or comprehension [13], they still have the advantage to reach out to a large sample of the general population in a short time with relatively low costs, without interfering with behaviors. Moreover, the structure of such questionnaires gives access to information regarding the different occupational (work, transport, household activities) and recreational domains that compose overall PA.

The Global Physical Activity Questionnaire (GPAQ) was developed in 2002 by the World Health Organization (WHO) as part of the STEPwise approach to survey chronic disease risk factor [14,15] and is now widely used for the national surveillance of PA in high- as well as in low/middle-income countries. According to the WHO, it has been administrated in more than 100 countries [16] and continues to be translated into and tested in many languages. The French version was recently validated against accelerometer measured PA and with an assessment of test–retest reliability [17]. The overall validity was poor to good but remained acceptable and similar to the previous study. GPAQ comprises 15 questions grouped to capture PA according to three behavioral domains (work, transport, and leisure), and one additional item collects the time spent sitting on a typical day [18].

As with other PAQs using a single item for sedentary time, the GPAQ measure of sitting time results in poor precision and low correlation with accelerometer-measured sedentary time [19,20], while PAQs using various sedentary behavior items have also shown only low-to-moderate correlations with accelerometer-derived sedentary time, total sitting time tends to be significantly underestimated by a single item compared to the total sitting time obtained with composite measures or accelerometer measurement [17,21,22,23,24]. Yet, the amount of time spent seated has become a matter of concern in public health as societal changes have made it the dominant posture in many situations of daily living. Moreover, sitting time is likely to continue increasing, although meta-analyses clearly point to its association with poorer health outcomes, cardiovascular diseases, and premature mortality [3,7,25,26].

The National Observatory for Physical Activity and Sedentariness (ONAPS) [27], with the expertise of the ONAPS scientific committee (http://www.onaps.fr (accessed on 19 May 2021)), has developed a new questionnaire aim at properly monitoring PA and SB, the ONAPS-PAQ. Inspired by the GPAQ concerning PA items, the ONAPS-PAQ includes detailed items when it comes to sedentary behaviors, such as sitting time at work, screen time at home, and time spent seated during transportation. It also captures time spent in household activities, whereas the GPAQ does not. The aim of the present work is to assess the validity and reliability (test–retest) of the ONAPS-PAQ when assessing PA and SB among adults.

## 2. Materials and Methods

### 2.1. Study Design and Participant Selection

The sample size was determined according to COSMIN recommendations with a minimum number of 100 subjects to evaluate concurrent and criterion validity and 50 participants to analyze reliability [28] appropriately. The study included adults aged from 18 to 69 years old, and subjects with cognitive impairment were excluded. Participants were recruited from faculties’ students, staff, and surrounding communities to participate in this pilot study. All subjects provided their informed consent for inclusion before they participated in the study. The study was conducted in accordance with the Declaration of Helsinki, and the protocol was approved by the Ethics Committee of CPP Sud Est VI (reference: 2020/CE 27).

During the first visit, participants completed both the ONAPS-PAQ and the GPAQ French version in a randomized order. Anthropometric measurements (height and weight) and body composition (Fat Mass and Fat-Free Mass) were assessed, and sociodemographic information was collected (age, sex, professional activity). They were then given an initialized accelerometer. Participants wore the accelerometer for the following seven days over the right hip during all waking hours and the non-dominant hand during sleeping hours. A valid day was defined as having 600 or more min of monitor wear. All recorded days in which total wear time was below this cut point were discarded from the analysis. For being considered a valid week, it was mandatory that the participant had at least 4 valid days, including one weekend day. A subsample of participants randomly selected using a random number table completed the ONAPS-PAQ twice, separated by a seven day-interval, to question its test–retest reliability.

### 2.2. Instruments

*GPAQ*. The GPAQ has been developed to assess PA by the WHO [15] and was validated in a French version [17]. It contains 16 items designed to assess the frequency and duration of PA in three domains, during work, transportation, and leisure time, at two levels of PA intensity, moderate (MPA) and vigorous (VPA). It reports PA duration by minutes·day^−1^ and days·week^−1^ for each PA domain and intensity level, which allows for calculating the energy expenditure scored in the metabolic equivalent of task (MET). One item reports the time spent sitting during a typical day. The GPAQ was used in its self-administrated form to test the ONAPS-PAQ concurrent validity.

*ONAPS-PAQ* is a self-administrated PA questionnaire developed by the ONAPS. It contains questions about PA in four main domains: at home, during work, during transport, and during leisure time. It derives from the GPAQ French version by including all of its items and additionally includes items from the International Physical Activity Questionnaire (IPAQ) long-form [29]: two items (minutes·day^−1^ and days·week^−1^) concerning household PA, details about active travel (days/week and minutes/day of walking, days/week and minutes/day of cycling), and details on SB (days·week^−1^ min·day^−1^ of travel by car, days·week^−1^ min·day^−1^ of sitting at work, days·week^−1^ min·day^−1^ of screen time at home and days·week^−1^ min·day^−1^ of other sitting times at home (see Appendix A).

*Accelerometer*. The UKK-RM42 accelerometer (UKK Terveyspalvelut Oy, Tampere, Finland) was used to record PA intensity, frequency, and duration. The size of the accelerometer is 35 × 27 × 9 mm, and the weight is 9.3 g. UKK RM42 is a triaxial accelerometer that was attached to an elastic belt on the right side of the hip at the level of the iliac crest during waking hours (excluding the time spent in the sauna, bath, shower, or in other water activities) and on the wrist band to non-dominant hand during the sleep time. The UKK RM42 was used along with a recent method to process the raw data. The mean amplitude deviation (MAD) is a method that can be applied to different hip-worn accelerometer brands collecting data in raw mode and has high validity for estimating PA intensities [30,31,32]. Further, the angle for postural estimation (APE) analysis was performed for differentiating stationary sitting and standing. Cut points for PA were defined [30,32]: PA was defined here as a MAD value above 22.5 mg (milligravity), which represents the cut point between stationary behavior and low-intensity PA (LPA). The stationary behavior was classified as standing if the APE was less than 11.6°, and otherwise as sitting or lying. Sitting and lying time only were considered as SB in the present study. The optimal MAD cut-point for 3.0 METs (separating MPA from LPA) was 91 mg and 414 mg for 6.0 METs (separating VPA from MPA). The data were analyzed in 6 s epochs and smoothed epoch-wise with a one-minute exponential moving average.

*Anthropometric measurement*. Body weight and height were recorded to the nearest 0.1 kg and 0.5 cm, respectively, while wearing light clothes and bare-footed, using a digital scale and a standard wall-mounted stadiometer, respectively. Body mass index (BMI; in kg m^−2^) was calculated as body weight divided by the square of height. Body composition was assessed on the same occasion using impedance analysis (Tanita MC 780). This Tanita MC780 device has been recently validated in young adults of various PA levels [33].

### 2.3. Scoring of Physical Activity and Data Reduction

Data from self-administrated questionnaires were cleaned for missing values and non-plausible responses according to the GPAQ Analysis Guide and its Cleaning Data with Epi-Info recommendations [18]. Concerning accelerometer data, a valid week consisted of at least 4 days worn with one weekend day per week, for a minimum of ten hours per day. If worn for 4–6 days, the days were averaged and multiplied by a respective factor to equal 1 full week.

PA data were used to compute estimates of total activity for each intensity or domain category captured by each instrument (GPAQ, ONAPS-PAQ, and accelerometer separately). (1) Self-reported PA times were calculated by multiplying the number of days in PA by minutes/day engaged at the level. ONAPS and GPAQ moderate and vigorous PA (MVPA) were calculated by adding the self-reported minutes of vigorous (VPA) and moderate (MPA) activity over the past seven days. VPA was calculated by the sum of VPA at work and VPA in leisure. MPA calculation refers to IPAQ guidelines in which walking time is weighted by 0.825, cycling time by 1.5, and housework time by 0.5 considering an energy expenditure of 3.3 metabolic equivalents of tasks (METs) for walking, 6 METs for cycling, and 2 METs for housework chores.
*VPA = VPA·work + VPA·leisure* (*min·week*^−1^)

*MPA = MPA·work + MPA·leisure + 0.825* * *walkingPA + 1.5* * *cyclingPA + 0.5* * *houseworkPA* (*min·week*^−1^)

*MVPA = MPA + VPA* (*min·week*^−1^)


The volume of the past seven days PA energy expenditure was estimated by multiplying minutes of moderate physical activity by 4 METs, multiplying minutes of vigorous physical activity by 8 MET, and adding both.
*Self-reported total PA = 4* * *MPA + 8* * *VPA MET·minute·week*^−1^

(2) Accelerometer data allowed additional measures of light physical activity (LPA). The median of light activities energy expenditure is 2.25 MET. The volume of total PA measured with the accelerometer was then calculated as the sum of light, moderate, and Vigorous PA-related energy expenditure.
*Accelerometer-measured total PA = 2.25* * *LPA + 4* * *MPA + 8* * *VPA MET·minute·week*^−1^

In regard to the WHO recommendations [34] and in regard to thresholds reported in epidemiological studies, participants were classified into three categories of PA: low level for MVPA < 150 min·week^−1^ (participants who do not meet WHO recommendations), moderate level for MVPA ∈(150;525) min·week^−1^, and high level for MVPA > 525 min·week^−1^ (90 min·day^−1^).

### 2.4. Statistical Analysis

Data analysis involved the use of R 4.0.2 (R Core Team, 2020. R Foundation for Statistical Computing, Vienna, Austria). Representativity of the reliability study subsample was tested using Student *t*-test and chi-2 on age, sex, and body mass index variables. For the full sample, Spearman correlation coefficients (ρ) were tested between PA parameters (MVPA, SB) and anthropometric parameters (BMI, Fat Mass). The PAQ tested in the present study is a patient-reported outcome measurement based on a formative model (items together form the construct: total PA) so that structural validity and internal consistency (Cronbach analysis) were not relevant. The categorical variables were reported as relative frequency or percentage and quantitative variables as median with the first and third quartile for dispersion parameters or mean and standard deviation for Gaussian variables. The Kolmogorov–Smirnov test was used to assess the normality of data distribution and showed that self-reported variables were not normally distributed with quite a few outpoints by far from the mean value. Reliability and validity analyses refer to COSMIN recommendations for patient-reported measurement tools [35].

*Test–retest reliability* was assessed on ONAPS-PAQ. Categorical variables, such as those with a “Yes/No” response (e.g., doing some vigorous activity at work), were assessed for reproducibility using the Cohen kappa coefficient and percent agreement [36]. Continuous variables, such as frequency (days) and duration (time in minutes) of PA and SB, were assessed for reliability within each domain (work, leisure, housework, transport) for each intensity (moderate, vigorous) using intraclass coefficient correlations (ICC) estimated with two-way random effects model [37]. According to Landis et al. ICC < 0.50 was considered as poor, between 0.50 and 0.75 as moderate, and >0.75 as good [38].

*Concurrent Validity.* Domains of PA and summary variables of total PA were computed from GPAQ and ONAPS-PAQ to assess their association using Spearman’s rho coefficients. For GPAQ, the following formulas were used: VPA = sum of vigorous PA at work plus vigorous PA in leisure domain; MPA = sum of moderate PA at work plus moderate PA in leisure domain plus total transport-related. For ONAPS-PAQ: moderate-intensity at work, in leisure domain, total transport and housework activities were combined to compute a total moderate-intensity variable, MPA (cf. formulas above). Sitting time refers to a single GPAQ item and four ONAPS-PAQ items (travel by car, sitting at work, screen time, and other sitting time at home), which were added to be compared.

*Criterion Validity.* The association between self-reported minutes of PA (ONAPS-PAQ, GPAQ) and minutes of PA measured by accelerometer per week was assessed using the Spearman correlation coefficient. Accelerometer data were aggregated to provide measures of average time spent in sedentary behaviors, light-, moderate- and vigorous-intensity activity. The correlation between these measures and self-reported PA (total PA, VPA, MPA, and MVPA) was assessed. In addition, the relationship between the computed variable ‘sitting time’ (the sum of daily car travel time, sitting time at home, and sitting time at work, a proxy measure of sedentary behavior) was compared with measures from the RM42 accelerometer. The following standards were applied to interpret the correlation coefficient: 0 to 0.2 = poor; 0.21 to 0.40 = fair; 0.41 to 0.60 = moderate/acceptable; 0.61 to 0.80 = substantial; 0.81 to 1.0 = near-perfect [38].

Both the concurrent and criterion validity of the ONAPS-PAQ were assessed by Bland–Altman plots to evaluate agreement and bias for MVPA and SB between the questionnaire’s answer and results from the accelerometer [39].

## 3. Results

*Characteristics of the population.* A total of 137 adults were recruited. None of them was lost to follow up. ONAPS-PAQ and GPAQ data regarding PA and SB were obtained for all subjects, but valid accelerometer measurements were obtained for 124 participants so that 13 subjects were discared from the analysis. The characteristics of these 124 participants are described in Table 1. The age range of the participants was 18–69 years, 32% were students, and 63% had a higher education qualification. The subsample participating in the test–retest protocol (reliability study, *n* = 37) was representative of the larger study population regarding age (39.7 ± 15.9 vs. 35.7 ± 13.7 years, mean ± SD, *p* > 0.05), sex distribution (61% vs. 63% of women), and body mass index (24.1 ± 4.9 vs. 23.2 ± 4.1 kg m^−2^), all *p*-values demonstrating no significant differences (*p* < 0.05). No significant correlation was found between PA and anthropometric parameters (BMI/MVPA ρ = 0.15, BMI/SB ρ = 0.07, FatMass/MVPA ρ = 0.09, FatMass/SB ρ = 0.10, all *p* > 0.05).

*Descriptive statistics of PA and SB*. Table 1 presents medians and quartiles for variables used to evaluate the validity of the ONAPS-PAQ showing the different PA and SB variables for ONAPS-PAQ, GPAQ, and accelerometer data. The number of minutes per week spent in MVPA was 2.6 times higher, estimated by the ONAPS-PAQ than by accelerometer data. VPA was particularly overestimated with self-reported answers. MPA was declared 2.2 times higher than accelerometer measures. Interestingly, most of the total PA energy expenditure was related to LPA according to accelerometer data and to MPA according to self-reported questionnaires data. The over-reported MVPA was larger with ONAPS-PAQ than GPAQ (1014 vs. 630 min·week^−1^), and the difference was related to a larger amount of MPA with the ONAPS-PAQ, whereas VPA was similarly reported in both questionnaires. There was little evidence of PA at work: VPA at work was reported in 9% of ONAPS-PAQ and 10% of GPAQ; MPA at work was reported in 50% of ONAPS-PAQ and 39% of GPAQ. Finally, weekly PA energy expenditure was essentially related to recreational PA (40%) and, to a lesser extent, to household chores (21%). Active transport (walking or cycling) comprised a small fraction (14%) of weekly MVPA. Concerning SB, the median-reported sitting time according to the ONAPS-PAQ was comparable to the median sitting time as measured by the accelerometer, with a larger dispersion of values, however (self-reported SB = 480 (360–652) min·day^−1^ vs. accelerometer-measured SB = 481 (432–566), the *median (q1*-*q3*)). In comparison, the GPAQ underestimated the median sitting time by 120 min·day^−1^ compared to the ONAPS-PAQ and accelerometer. By categorizing level of PA (low level: participants under 150 min·week^−1^ of MVPA, the minimum recommended for health; moderate and high level of PA: MVPA threshold of 600 min·week^−1^), the ONAPS-PAQ showed a majority of participants with a high level of PA, whereas accelerometry showed a large majority of participants with moderate levels of PA. However, the proportion of low active participants was comparable with ONAPS-PAQ and accelerometry.

*Test–retest reliability.* Intraclass correlation coefficients (ICC) for each activity level and activity domains are provided in Table 2. With a 7-day interval, ICC ranged from 0.71 to 0.99 (*n* = 36). All ICC were good except for sitting time in car travels which showed moderate ICC (0.71 CI_95_ (0.49-0.84)). Categorical variables concerning physical activity and sedentary behaviors (yes/no) all showed good Cohen’s kappa correlations.

*Concurrent validity.* Spearman’s rho correlations coefficient presented in Table 3 showed substantial to near-perfect correlations between ONAPS-PAQ and GPAQ, ranging from ρ = 0.66 to ρ = 0.86 except for MPA in leisure time which showed acceptable correlation (ρ = 0.56, *p* < 0.05). The highest correlations were found in VPA, particularly in leisure domain (ρ = 0.86, *p* < 0.05) and total PA (ρ = 0.77). Correlations were slightly lower for MPA ranging from ρ = 0.56 to ρ = 0.71, transport and SB (ρ = 0.66 and 0.64, respectively, both *p* < 0.05). Results of Bland–Altman analysis for MVPA (Figure 1A) demonstrated a mean difference (ONAPS-PAQ minus GPAQ) of +378 _IC95_(246-510) min·week^−1^. Limits of agreement for the two instruments were wide, with the difference lying between −1018 min·week^−1^ and +1775 min·week^−1^. Following a review of Figure 1A, it would appear that the mean difference was still comparable, whatever the level of MVPA reported. The Bland–Altman analysis of SB (Figure 1C) showed a mean difference of +98 _IC95_(59-138) min·day^−1^ with wide limits of agreement (from −318 to 514 min·day^−1^) but comparable whatever the sitting time reported. In both, the Bland–Altman plot showed a heterogeneous bias between the GPAQ and ONAPS-PAQ with homogeneous error variation (Figure 1A,C).

*Criterion validity.* Positive correlations were observed between ONAPS-PAQ and accelerometer PA variables (Table 3, ρ = 0.26–0.39). The lowest Spearman’s rho was found for MPA and the highest for total PA. A result not reported in Table 3 showed a higher correlation between ONAPS-PAQ MPA and accelerometer-measured LPA (ρ = 0.39, *p* < 0.05) than accelerometer-measured MPA. SB reported in ONAPS-PAQ was also moderately correlated to accelerometer measure (ρ = 0.41, *p* < 0.05), better than GPAQ correlation with accelerometer SB (ρ = 0.25, *p* < 0.05). Results of the Bland–Altman analysis for MVPA (Figure 1B) demonstrated a mean difference (ONAPS-PAQ minus Accelerometer) of +838 _IC95_(663;1014) min·week^−1^. Limits of agreement for the two instruments were wide, with the difference lying between −1020 min·week^−1^ and 2697 min·week^−1^. Following a review of Figure 1B, it would appear that a positive bias existed for the ONAPS-PAQ. The people who were more physically active were found to be more likely to over-report their level of physical activity using the ONAPS-PAQ. The Bland–Altman analysis of SB comparing ONAPS-PAQ to accelerometer (Figure 1D) showed a small mean difference of +9.9 _IC95_(−25.1;44.9) min·day^−1^ with wide limits of agreement (from −361 to 381 min·day^−1^). Following a review of Figure 1D, it would appear that a slight bias existed for the ONAPS-PAQ. The people who were found to be more sedentary were more likely to over-report their level of SB using the ONAPS-PAQ, and conversely, the people who were less sedentary were found to be more likely to under-report their level of SB (Figure 1D).

## 4. Discussion

There is today a need for easy-to-use methods to properly assess PA and SB at a population level; the ONAPS-PAQ showed modest and good validity against objective (accelerometer) and subjective (GPAQ) measures of PA, respectively. It also showed a better estimation and correlation of sitting time than GPAQ when compared to accelerometer measurement.

The epidemiology of sedentariness and physical inactivity, both known as morbidity-mortality risk factors, largely and increasingly involved in the current burden of non-communicable diseases, needs harmonized and validated measurement instruments for population studies [40]. The GPAQ is largely used by many countries’ epidemiological monitoring systems. However, it was not constructed for the assessment of sedentary behavior (SB) and failed to estimate sitting time adequately. The ONAPS-PAQ is a PAQ developed by the French national observatory for physical activity and sedentariness. It was derived from the GPAQ, including the evaluation of housework-related PA and different sedentary behaviors, such as car travel, screen time, and other sitting occasions at home. The present test–retest evaluation of the ONAPS-PAQ revealed a good reliability of the questionnaire, and the concurrent validity study with GPAQ showed substantial to high agreement coefficients. The criterion validity study, comparing ONAPS-PAQ to accelerometer measures, showed fair agreement coefficients of the same order as those usually reported in PAQ validity studies [17,29,41,42,43].

The correlation obtained regarding sedentary time between ONAPS-PAQ and accelerometer was amongst the highest in the available literature [41,43]. SB is known to be regularly and substantially underestimated by self-reported PAQ. Bauman et al. describing the epidemiology of sitting in 20 countries using the IPAQ and Bennie et al. using the Eurobarometer survey results in 20 European countries, both found an average sedentary time of 5 h a day [44,45], whereas the meta-analysis from Stamatakis et al., based on accelerometer measurements, reported 8 h a day [3]. The present study shows that by incorporating more detailed items regarding sedentary behaviors, the ONAPS-PAQ seemed to provide a deeper and more complete evaluation of sedentary time compared to GPAQ. It showed indeed a mean daily sitting time significantly higher than the GPAQ one and very close to the median obtained with accelerometer measures, with greater dispersion parameters, however. Lastly, information on sedentary behaviors provided by the ONAPS-PAQ may be of interest for epidemiological studies. The results obtained in the present sample reported that most of the daily sedentary time was explained by sitting at work (41%) and screen time at home (30%), the latter being recognized as the main determinant for the morbidity of sedentariness [46,47] while generally not provided by the other PAQs. Precautions have to be taken when interpreting the present results, particularly since sedentary time is known difficult to self-report, even using questionnaires particularly focusing on sedentary behaviors [48].

The concurrent validity of the ONAPS-PAQ was assessed using GPAQ, an instrument designed with the same purpose of population monitoring results from the two being strongly correlated. The strongest correlations were observed for vigorous physical activities, as regularly observed [14,17,49,50]. Correlation coefficients were slightly higher than those usually found in the literature, which is not surprising considering the similarity of the PA items in both questionnaires.

Results of the criterion validity study were similar to previous studies assessing the validity of PAQ against device-based measurements showing poor to fair correlations. It is well recognized that self-reported measures overestimate MVPA [51]. The present results showed an overestimation of approximately two to three times the amount of moderate activity that was recorded on the accelerometer and overestimated the amount of vigorous activity. These concerns about overestimation are similar to those reported in previous studies that have identified an overreporting of MVPA with GPAQ [17,50], and even more significant with the IPAQ long-form (IPAQ-LF). Riviere et al., studying concurrent validity of a French version of the GPAQ to the IPAQ-LF, showed that the latter reported 637 ± 1641 MET min·week^−1^ more than GPAQ measures for total PA. A possible explanation advanced was that IPAQ-LF contains detailed items dedicated to household activities, whereas GPAQ did not. Such items were also included in the ONAPS-PAQ, which could explain the difference observed for total PA between ONAPS-PAQ and GPAQ. Another interesting point was that self-reported MPA correlated better with accelerometer-measured LPA. Similar results have been found in validation studies of the IPAQ and GPAQ [42,52]. This may reflect the problem of estimating the intensity of PA.

The test–retest analyses showed good reliability of the ONAPS-PAQ. With a seven-day interval repetition, high intraclass correlation coefficients were obtained in all PA domains, all PA categories, and sedentary times (ρ = 0.77–0.99). Such noteworthy results need, however, to be tempered. Indeed, the number of participants did not reach the expected fifty subjects recommended in COSMIN guidelines for self-reported outcome measurement instrument reliability [35]. Furthermore, a memory bias with a seven-day interval between two questionnaire completions, although not unusual, may not be excluded. Concerning the reliability of the SB items, the test–retest analysis also showed high ICC (*n* = 36, ρ = 0.71–0.9). These results are consistent with the general good reliability of PAQ.

This study presented several strengths, notably with the concordant measurement period for both questionnaires (the same seven days) and the use of the accelerometer for criterion validity. Indeed, the accelerometer is currently recognized as a reliable alternative to measuring PA objectively in daily life [53]. In addition, the Bland–Altman analysis we used is now a recommended approach to assess the level of agreement compared with correlation coefficients assessing only the strength of the relationship between the measures [35,39,41]. The major limitations of the study were the lack of a long-term reliability assessment and the number of participants to the reliability study, which did not meet COSMIN criteria. Another limitation was that the sample included too few inactive volunteers to be compared to other groups, and young adults, girls, students, and high educational level were over-represented. Therefore, it is recommended that further investigations be conducted using ONAPS-PAQ on larger samples as well as targeting specific populations and that sensitivity of the ONAPS-PAQ to behavioral changes should be explored.

## 5. Conclusions

The present validity and reliability study suggested that the ONAPS-PAQ is a reliable questionnaire for use in the adult population. The overall validity remained acceptable and was similar to previous studies assessing the validity of the PAQs currently used in epidemiological monitoring systems [41,54,55]. As it was derived from the GPAQ, its data could be compared and aggregated to other international epidemiological studies. The strength of the ONAPS-PAQ was its ability to investigate SB. Criterion validity of SB measurement was higher than GPAQ, the measurement bias was significantly reduced, and the sedentary time estimation similar to accelerometer measurements. Regarding PA, MVPA was largely overestimated compared to accelerometer measurement. Although the attempt to quantify household PA is interesting for research purposes, related items of the ONAPS-PAQ could have introduced further overestimation of the total PA [17,55]. Adjustments in education or instructions on what is meant by moderate PA and vigorous PA should also be considered. Lastly, the proportion of inactive people appears comparable when assessed by the ONAPS-PAQ and accelerometry, but there were too few inactive subjects in the sample to support statistical analyses. Further studies on a larger sample, specific populations, and sensitivity to behavioral changes should be conducted.

## Figures and Tables

**Figure 1 ijerph-18-05643-f001:**
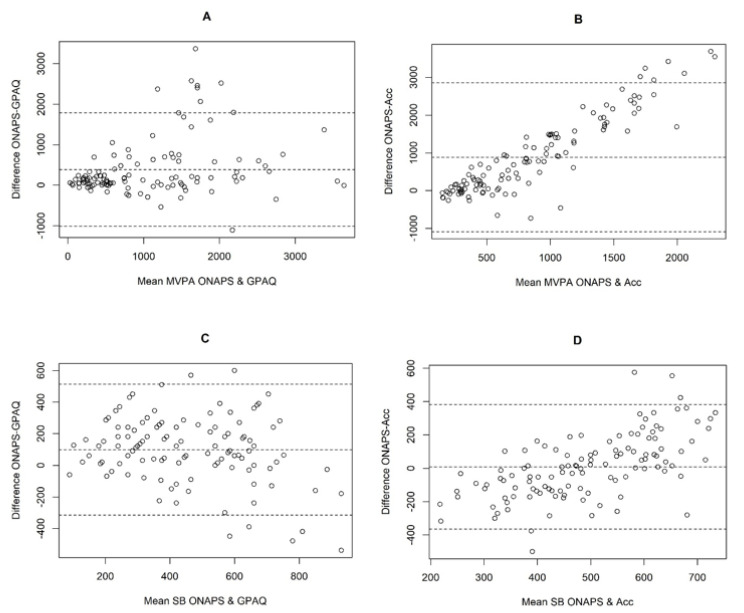
Bland–Altman plots for the agreement of data assessed with ONAPS-PAQ, GPAQ, and accelerometer (Acc). MVPA, moderate-to-vigorous physical activity (min·week^−1^); SB = sedentary behavior (min·day^−1^). (**A**,**C**): agreement of ONAPS-PAQ with GPAQ for MVPA and SB; (**B**,**D**): agreement of ONAPS-PAQ with RM42 accelerometer for MVPA and SB. ONAPS-PAQ = national observatory of physical activity and sedentariness-physical activity questionnaire; GPAQ = global physical activity questionnaire.

**Table 1 ijerph-18-05643-t001:** Demographic and anthropometric characteristics for all participants, men and women (mean ± sd% in each category of age and body mass index) and data for physical activity and sedentary times (*median* (*1st*-*3rd*) quartiles) measured by ONAPS Physical Activity Questionnaire (ONAPS-PAQ), Global Physical Activity Questionnaire (GPAQ), and RM42 accelerometer (Acc).

**Demographic and Anthropometric Characteristics**			
	Total	Men	Women
N	124	46	78
Sex (%)		37	63
Mean Age(years)	35.7 ± 13.7	35.2 ± 14.0	36.1 ± 13.5
Age Category (%)			
18–39 years	65	64	65
40–59 years	28	29	27
≥60 years	7	7	8
BMI (kg m^−2^)	23.2 ± 4.1	24.0 ± 3.4	22.7 ± 4.4
BMI category (%)			
(20–25)	56	61	53
(25–30)	17	24	13
>30	6	6	6
Fat Mass (%) Fat Free Mass (kg)	22.7 ± 9.0 49.8 ± 10.9	17.1 ± 7.1 61.2 ± 6.1	25.6 ± 8.5 43.9 ± 7.8
**Physical Activity and Sedentary Behaviors Measurement**			
	*Accelerometer*	*ONAPS-PAQ*	*GPAQ*
Total PA (MET·min·week^−1^)	5846 (4480–6817)	5171 (2604–10,305)	3260 (1710–6960)
Sitting time (min·day^−1^)	481 (432–566)	480 (360–652)	360 (240–540)
PA by intensity (min·week^−1^)			
Light	1708 (1426–2105)	n/a	n/a
Moderate	359 (233–501)	806 (264–1499)	400 (120–1030)
Vigorous	5 (1–51)	150 (0–360)	150 (0–360)
MVPA	388 (257–526)	1014 (472–2040)	630 (310–1345)
PA by domain			
Work			
Moderate	n/a	15 (0–1057)	0 (0–600)
Vigorous	n/a	0 (0–0)	0 (0–0)
Leisure			
Moderate	n/a	120 (56–240)	90 (0–240)
Vigorous	n/a	120 (0–270)	120 (0–270)
Transport	n/a	86 (0–217)	80 (0–210)
Housework	n/a	105 (45-208)	n/a
PA level (%)			
Low	5	4	10
Moderate	69	24	33
High	26	72	57

Abbreviations: PA = physical activity; MVPA = moderate and Vigorous PA; n/a = not assessed; MET = metabolic equivalent of task.

**Table 2 ijerph-18-05643-t002:** Test–retest reliability of the ONAPS-PAQ by activity intensity and physical activity domains with two repetitions (36 participants). Intraclass correlation coefficients (ICC) for the quantitative physical activity measures comparison and Cohen kappa coefficient for personal concern on practices (yes/no) are reported.

	7 Day-Interval Test Repetition (*n* = 36)			
	Test1	Test2	ICC (95% CI)	Cohen Kappa
PA category (min·day^−1^)				
Moderate	87 ± 85	75 ± 68	0.96 (0.91–0.98)	
Vigorous	34 ± 33	35 ± 34	0.9 (0.81–0.95)	
MVPA	122 ± 91	110 ± 79	0.95 (0.91–0.98)	
PA domain (min·day^−1^)				
Work moderate	42 ± 73	33 ± 56	0.98 (0.96–0.99)	0.93
Work Vigorous	4 ± 13	4 ± 14	0.99 (0.97–1)	0.99
Leisure moderate	26 ± 29	26 ± 31	0.92 (0.856–0.96)	0.61
Leisure Vigorous	30 ± 31	30 ± 28	0.87 (0.76–0.93)	0.78
Transport	19 ± 23	23 ± 30	0.77 (0.59–0.88)	0.92
Housework	33 ± 52	40 ± 43	0.91 (0.83–0.95)	0.9
Sitting time (min·day^−1^)				
Work	255 ± 208	293 ± 235	0.9 (0.826–0.95)	0.98
Car travel	41 ± 32	51 ± 38	0.71 (0.49–0.84)	0.85
Home screen	156 ± 116	172 ± 107	0.79 (0.63–0.89)	0.99
Others home sitting	96 ± 84	87 ± 66	0.77 (0.59–0.87)	0.77
Total sedentary time	548 ± 259	603 ± 283	0.88 (0.77–0.94)	

Abbreviation: PA = physical activity; ICC = intraclass correlation coefficient; 95% CI = 95% confidence interval.

**Table 3 ijerph-18-05643-t003:** Results of Spearman correlations between accelerometer (Acc) and ONAPS-PAQ (Spearman’s rho Acc, criterion validity) measure, between accelerometer and GPAQ measure, and between ONAPS-PAQ and GPAQ measure (Spearman’s rho GPAQ, concurrent validity).

	Rho Acc/ONAPS	Rho Acc/GPAQ	Rho ONAPS/GPAQ
Total PA (MET·min·week^−1^)	0.39 *	0.35 *	0.77 *
Sitting time (min·day^−1^)	0.41 *	0.26 *	0.64 *
PA by intensity (min·week^−1^)			
Moderate	0.26 *	0.30 *	0.71 *
Vigorous	0.31 *	0.38 *	0.80 *
MVPA	0.34 *	0.41 *	0.76 *
PA by domain			
Work			
Moderate			0.66 *
Vigorous			0.73 *
Leisure			
Moderate			0.56 *
Vigorous			0.86 *
Transport			0.66 *

* *p* < 0.05 Abbreviation: PA = physical activity; MVPA = moderate and Vigorous PA; n/a = not assessed; MET = metobolic equivalent of task.

## Data Availability

The data presented in this study are available on request from the corresponding author. The data are not publicly available due to nominative datasets that must be protected in accordance with the General Data Protection Regulation.

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
