# Peer review of "Reliability and Validity of the ONAPS Physical Activity Questionnaire in Assessing Physical Activity and Sedentary Behavior in French Adults"

_ijerph, 2021, doi:10.3390/ijerph18115643_

Round 1
Reviewer 1 Report
Original submission: Reliability and Validity of the ONAPS Physical Activity Questionnaire in Assessing Physical Activity and Sedentary Behavior in French Adults.
Comments to the authors:
The authors assessed the concurrent and criterion validity of the ONAPS-PAQ in 137 healthy adults as well as its test-retest reliability in a subgroup of 36 participants. The study found that the ONAPS-PAQ had good reliability and acceptable validity. It also suggested that the ONAPS-PAQ assessed sedentary behavior better than the Global Physical Activity Questionnaire. Overall, the authors have done a nice job describing the study and procedures. I have a couple comments and questions below:
Comments:
- Physical activity questionnaires are abbreviated (PAQs) here (Line 46) so please makes sure it is throughout the manuscript.
- Revise “As others…” to “As other…” (Line 65).
- “…monitor physical activity and sedentary”. Add behavior after “sedentary”? (Line 78)
- Were there any inclusion/exclusion criteria for this study? If yes please include (Section 2.1).
- Abbreviate “Global Physical Activity Questionnaire” (Line 104).
- The National Observatory of Physical Activity and Sedentariness should be abbreviated here (Line 112).
- Please describe, under the Methods section, how participants returned the accelerometers.
- Double check “et chi-2”. Not sure what “et” means (Line 186).
- Please revise “sexe” to “sex” (Lines 186 and 236).
- The sample sizes reported in the Results section are different from the ones described in Abstract and Methods. Please check which are correct. Also, please include description if there were participants lost to follow up.
- Following the above comment, did all participants return the accelerometer? Was there data missingness that needed to be addressed?
- Please check all terms that have been abbreviated and make sure they are consistent throughout the manuscript.
- Double check the wording of this sentence (Lines 383-384).
- What was the rationale for having another subgroup study for the 9 participants completing the 8-day accelerometer-wearing? (Line 422)
- Follow up on the previous question. Is there a reason why the data was not incorporated into the current analysis?
- I would recommend the manuscript undergo a careful editing as well as grammar/typo check if invited for next submission.
Reviewer 2 Report
This paper discussed the reliability and validity of the ONAPS-PAQ, which is helpful in better getting the personal PA and SB. However, this manuscript is not well written, and a major revision is needed before its publication.
- The English writing need a careful check. Too many errors and incomplete sentences existed in the paper, even in the first sentence of this paper.
- It is better to cite the website through the reference form, which lists the websites in the reference list.
- The producer of used accelerometer is mentioned twice, but the information of this sensor is not mentioned at all, e.g. size, weight, comfortability, etc. They may change the activities of volunteers, if the sensor is not comfortable enough. A discussion is needed.
4. More results can be obtained and analyzed. Are there correlations between PA / SB and BMI (or weight), age, female/male, and even with/without high education (also mentioned in the paper). Do validity results vary with these parameters?
Reviewer 3 Report
To authors
This manuscript is for verifying a new test method (ONAPS-PAQ) that measures PA and SB with a questionnaire.
If you modify the content of this manuscript in consideration of the following points, it will be even better.
Major revision
L.449-450
This conclusion should be rewritten to include L.457-459.
Because the ONAPS-PAQ was excellent only in terms of SB. And the purpose of this study was to evaluate the ONAPS-PAQ in terms of PA and SB.
Minor revision
1 Table 1: Sitting time(min/day)
Not 481(233; 501), correct 481(432; 566)?
2 Please give a supplementary explanation about the following numerical values.
L.303
0.66-0.86
L.305
0.73-0.86
L.306
0.56-0.71
3 L.327
Not r = 2.25, correct r = 0.26?
Reviewer 4 Report
This manuscript presents findings linked to the validity and reliability of a new national-level PAQ, namely the ONAPS, developed through the French National Observatory for Physical Activity and Sedentary Life. The paper is detailed and articulates a rationale around the need for a non-objective tool to better-assess composite PA - and specifically sedentary behaviours known to be underestimated using other self-report tools. The new tool also includes PA domains not present in some other instruments (e.g. household activities). It is really good to see the development of this national-level tool, for use with French adults and I think it will contribute to an important an ongoing debate around measurement of 24 hour physical activity (including sed and sleep time). I enjoyed reading the manuscript.
Intro:
Check phrasing line 44
Line 52 – suggest ‘interfering with behaviours’.
Line 53 – questionnaires instead of questionnaire?
Line 54 comprise or compose?
Line 55 – was instead of has been?
Suggest a thorough proof read of English language throughout the manuscript to address minor points (I won’t list further as they really are minor and would also probably be picked up by the copy editor).
Line 62 – Useful point made about validation of French GPAQ with accelerometery-derived PA – I think it would be helpful to include a short sentence about the findings.
Methods:
Line 89 – appreciate healthy participants were recruited, but were there any exclusion criteria? Thinking about application to other populations, particularly given the limitations as stated from line 444.
Typo line 92.
Please provide information on how the questionnaires were administered. For example were Show cards used?
Line 96 – suggest removing Tanita info here as replicated later in the methods.
Line 122 onwards – please refer to accelerometery procedures in the past tense throughout.
Refs:
Check all are fully presented (e.g. missing detail ref 27).
Line 178 – mindful of 2020 WHO guidelines on PA and sed behaviour https://bjsm.bmj.com/content/54/24/1451 – it would be prudent to note somewhere in the manuscript how these may impact on the processes employed here, indeed if the authors consider that they might?
Table 1: For readability, would this be better split into two tables, one for each of Sample Characteristics and a further for PA and sed behaviours measured? I would also suggest the title for table 1 reads as the accompanying text ‘Demographic and anthropometric characteristics’. Some formatting/typo issues to address.
Line 273 – were these truly qualitative variables?
Discussion: This is detailed and I have no major suggestions to make. Minor point re line 395 – suggest not referring to ‘near perfect’ correlations as perhaps arguably they are not.
Line 423 – check phrasing.
Line 444 onwards – regarding limitations and recommendations for further work, just a suggestion: I think there is an opportunity to really spell out clearly what might be done next with this tool, framing this study as the first to test the questionnaire.
Conclusions:
Lines 449-50: I think the opening sentence about reliability might be more cautious, given the low n?
Similar to my previous point, I think there is an opportunity here to firmly state what value this work brings – for example, might French researchers with a particular interest in self-report assessment of sed behaviours wish to consider this in place of other tools? Can you be more clear about whether the tool needs further adaptation before recommending it’s use (e.g. around the inclusion of the additional questions)?
Refs:
Check all are fully presented (e.g. missing detail ref 27).
Round 2
Reviewer 2 Report
Thank for taking my comments, and the revisions is good.
Reviewer 3 Report
This new version represents a massive effort and it deserves to be published.